# Stress Associated with Simulated Transport, Changes Serum Biochemistry, Postmortem Muscle Metabolism, and Meat Quality of Broilers

**DOI:** 10.3390/ani10081442

**Published:** 2020-08-18

**Authors:** Aijuan Zheng, Shumei Lin, Shoaib Ahmed Pirzado, Zhimin Chen, Wenhuan Chang, Huiyi Cai, Guohua Liu

**Affiliations:** 1Key Laboratory for Feed Biotechnology of the Ministry of Agriculture and Rural Affairs, Institute of Feed Research, Chinese Academy of Agriculture Sciences, No.12 Zhongguancun South Street, Haidian District, Beijing 100081, China; zhengaijuan@caas.cn (A.Z.); dr.pirzado@gmail.com (S.A.P.); chenzhimin@caas.cn (Z.C.); changwenhuan@caas.cn (W.C.); caihuiyi@caas.cn (H.C.); 2Veterinary Medicine Department, College of Animal Science and Veterinary Medicine, Shenyang Agriculture University, No.120 Dongling Road, Shenhe District, Shenyang 110866, China; 2004500035@syau.edu.cn

**Keywords:** broiler, meat quality, motion, vibration, muscle metabolism, transportation stress

## Abstract

**Simple Summary:**

The objective of this study was to investigate the effects of the transportation on broilers through the use of a vibration and motion simulation machine. Short-term transport stress induced significant physiological changes in broiler chickens. Simulation of the vibration and motion associated transport for 2 h altered hormonal secretion and blood biochemical characteristics. However, after 4 h, the birds appear to regain homeostatic equilibrium. Despite this, the stress activated antioxidant defenses, acidified muscles and increased peroxidation, as well as decreasing the meat quality of broilers. This study supports and extends previous work that identified transportation as a major risk in relation to bird welfare and meat quality.

**Abstract:**

Many factors contribute to the stress of transporting broilers from the farm to the processing plant. Using a motion simulation machine, a total of 144 male broilers were employed to determine the effect of motion, vibration, and feed withdrawal during transportation on serum biochemical parameters, postmortem muscle metabolism, and meat quality of broilers. The results indicated that transportation did not affect the activity of lactate dehydrogenase, γ-glutamyl transferase, aspartate aminotransferase, creatine kinase, and glucose in the serum, glutathione peroxidase in the breast and thigh muscle, nitric oxide synthase (NOS) in the breast, and heat stress protein 70 mRNA expression level in the liver (*p* > 0.05). Serum triiodothyronine, thyroxine, and insulin concentration declined with 2 h transportation (*p* < 0.05) and recovered with 4 h transportation (*p* < 0.05). NOS concentration in the thigh increased with 2 h transportation (*p* < 0.05) and recovered with 4 h transportation (*p* < 0.05). Two-hour and 4 h transportation increased the activity of superoxide dismutase in both muscles. Malondialdehyde, lactic acid, and drip loss_24 h_ in both thigh and breast muscles increased, and glycogen in both muscles decreased with increasing transportation times (*p* < 0.05). Two-hour transportation did not influence pH_45 min_ and pH_24 h_ in the breast and thigh muscle, but these indexes decreased with 4 h transportation. This experiment supports and extends previous work that identified transportation as a major risk in relation to bird welfare and meat quality.

## 1. Introduction

Transport from the farm to the processing plant imposes a stress on the animal [1]. This may result in reduced animal welfare and economic loss through downgrading of carcasses [2]. The physiological, biochemical, hormonal, and immunological effects of transportation have been documented in chickens [3,4,5]. During transit, chickens are exposed to numerous potential stressors, including handling, feed withdrawal, noise, vibration, thermal extremes, social disruption, crowding, and restriction of movement [6]. In broilers, transportation results in increased concentrations of corticosterone [7,8], changes in energy and protein metabolism [9], and an immunological challenge [10]. Transportation, combined with feed withdrawal and catching, altered the expression of hepatic genes associated with nutrient metabolism, cellular control, and immune function of broiler chickens [11]. Moreover, transport stress also results in higher chicken mortality and live weight loss [12,13]. Transportation stress increases endogenous microbial contamination of broilers, and this aggravates the microbial risk for the consumer of poultry meat [14].

Transport-induced stress may occur as a result of a combination of factors, such as vibration and motion, noise, stocking density and crowding, mixing of unfamiliar groups, and temperature fluctuations, together with feed and water deprivation [15]. In an attempt to investigate the effect of these factors, a transport simulator was reported. The simulation model not only simulates different parameters of actual transportation, but also allows control of experimental repeatability. Two different levels of vibration for four intervals of time (0, 30, 60, and 90 min) were subjected to simulate the transportation using mechanical simulator, which did not affect chicken rectal temperature and weight loss [16]. Moreover, increasing vibration duration elevated the total white blood cell count, and heterophil-to-lymphocyte ratio of the hens. Simulated transportation did not affect egg production and plasma protein, but had an effect on the hens’ feed intake and blood glucose level [17]. Most of the studies investigated the effects of actual road transport on broilers [18,19,20], while there are few studies on the effects of simulated transport on the meat quality of pre-slaughter broilers. Therefore, the objective of this study was to investigate the effects of the stress associated with the vibration and motion aspects of transport on serum biochemistry, postmortem muscle metabolism, and meat quality of broiler chickens.

## 2. Materials and Methods

### 2.1. Ethics Statement

Feeding trials were conducted according to the guidelines for animal experiments set out by the National Institute of Animal Health, and all animal procedures were approved by the Chinese Academy of Agricultural Sciences (statement no. AEC-CAAS-20191106).

### 2.2. Bird Management and Experimental Design

One-day-old, Arbor Acres (AA) male chicks were obtained from a commercial hatchery (Huadu Foodstuff, Beijing, China) and were kept in a temperature-controlled room. The birds were housed in the Nankou experimental farm of the Feed Research Institute, CAAS, Beijing, China. Birds were raised in accordance with the AA Broiler Management Guide. Chicks were vaccinated for Marek’s Disease at 1 day-old and for Newcastle Disease and Infectious Bronchitis at 7 days post-hatching. Room temperature was maintained at 33 °C for days 0–3 and gradually reduced to 24 °C and maintained at 24 °C till the end of the study. Photoperiod was controlled to 16 h of light and 8 h of darkness. Relative humidity was set at 60–70% during the first week and then at 50–60% for the rest of the experiment. The stocking density was 8 birds/m^2^ in the cage.

Feed and water were provided ad libitum, and the diet formulations are listed in Table 1. At 39 d of age, 144 broilers of similar bodyweight were randomly allocated to three treatment groups, each of which had 6 replicates, with 8 birds per replicate. The same person handled chickens in the same way and put them in the cage. Chickens were housed in cages, with a density of 8 birds/0.64 m^2^. The whole cages (0.64 m^2^) were subjected to the transport simulation. The treatments were: (1) No transport but feed and water deprivation for 2 h; (2) Simulated transport with feed and water deprivation for 2 h; and (3) Simulated transport with feed and water deprivation for 4 h. A transportation simulator was used to simulate the vibration and motion associated with road transport. The Transportation Simulator (BF-SV, Bell Test Instrument Company, Dongguan, China) was used to simulate transporting road conditions, and the transportation vibration rate was set at 100 rpm.

### 2.3. Sample Collection

Three birds of each replicate were randomly selected and weighed. Blood was collected by cardiac puncture using heparinized tubes. The serum was obtained by centrifugation at 3000× *g* at 4 °C for 5 min. The serum samples were frozen immediately in liquid nitrogen and stored at −80 °C for the determination of serum parameters. One bird of each replicate was then euthanatized after electrical stunning [5]. Muscles of the breast and thigh were removed immediately and evaluated for pH (three different positions each bird) and drip loss (three pieces each bird), as described previously with modifications [21]. Three small pieces of breast muscle, leg muscle, and liver of each bird were taken and frozen immediately in liquid nitrogen for biochemical determination.

### 2.4. Parameter Determination

Serum triiodothyronine (T_3_), thyroxine (T_4_), and insulin concentrations were determined by radioimmunoassay (RIA; Diagnostic Products Corp., Los Angeles, CA, USA). Serum lactate dehydrogenase (LDH), γ-glutamyl transferase (GGT), asparate aminotransferase (AST), and creatine kinase (CK) were analyzed by enzyme immunoassay (EIA, Audit Diagnostics Ltd., Cork, Ireland). Serum glucose and muscle glucogen, glutathione peroxidase (GSH-Px), superoxide dismutase (SOD), nitric oxide synthase (NOS), lactic acid (LA), and malondialdehyde (MDA) concentrations were determined with commercially available kits (Nanjing Jiancheng Bioengineering Institute, Nanjing, China) accoring to the manufacturer’s instructions.

### 2.5. RNA Extraction, Reverse Transcription, and Real-Time Quantitative PCR

Specific primers (Table 2) suited to simultaneously amplify the HSP70 gene were designed according to the corresponding gene sequences in the GenBank library by using primer design software (Beacon Designer 7.51, PREMIER Biosoft International Palo Alto, San Francisco, CA, USA). Total RNAs were prepared from the liver using TRIzol regent (Takara bio, Shiga, Japan), and cDNA synthesis was performed using TaKaRa RNA PCR Kit (AMV) Ver.3.0 (Takara bio, Shiga, Japan), according to the manufacturer’s instructions. Real-time PCR was conducted using an iQ5 Multicolor Real-Time PCR Detection System (Bio-Rad, Hercules, CA, USA). PCR was performed in 20 μL reaction system containing 0.5 μL cDNA, 5 pmol forward and reverse primers, 10 μL SYBR Green Supermix, and water. Fold-change was calculated using the 2^−ΔΔCt^ method [22].

### 2.6. Statistical Analysis

Outliers were determined as values that deviated from the treatment mean by more than 1.5 times the interquartile range [23]. No outliers were identified and removed from the data set. The normality of variables was tested using the Explore procedure. The variance of INS was non-normal, and the transformation of ARSIN (SQRT(INS/10)) was used. Differences in the determined values among groups were analyzed with a one-way ANOVA model (SPSS Version 16.0, SPSS Inc., Chicago, IL, USA) and significant effects were further investigated with Duncan’s multiple range analysis. The homogeneity of the variance among treatments was performed. In the ANOVA models, transportation condition was the main effect. Results were presented as the mean and pooled SE. An error probability of *p* < 0.05 was considered to be statistically significant.

## 3. Results

### 3.1. Blood Hormones and Metabolites

The effect of different periods of exposure to motion stress on blood hormones and metabolites is presented in Figure 1A,B. Serum T_3_ (*p* = 0.034), T_4_ (*p* = 0.026), and insulin (*p* = 0.042) concentrations declined after 2 h of simulated transport, but recovered as the time of transportation increased (Figure 1A). Serum glucose and LDH, GGT, AST, and CK were not significantly affected by the stress (*p* > 0.05). Transportation tended to increase the concentration of serum GGT, AST, and CK (Figure 1B), but decreased the concentration of serum glucose (Figure 1A).

### 3.2. Meat Quality Characteristics

The effect of transportation stress on drip loss and pH is presented in Table 3. Transport affected pH and drip loss of breast and thigh muscle (*p* < 0.05). The drip loss_24 h_ in the breast and thigh muscle increased as the transport time increased (*p* < 0.05). The thigh and breast muscles pH_24 h_ and pH_45 min_ decreased after 4 h transport (*p* < 0.05), but these values did not change after 2 h transport (*p* > 0.05).

### 3.3. Activities of Key Enzymes from Breast and Thigh Muscle

Transportation stress did not affect the activities of GSH in both breast and thigh muscle significantly (*p* > 0.05). The activity of SOD in both muscles were increased by 2 h transportation (*p*_breast_ = 0.031, *p*_thigh_ = 0.028), and tended to recover after 4 h transportation (Figure 2), but these values were still higher than that of the control group (*p* < 0.05). There was no significant difference in the level of NOS in breast muscle between groups (*p* > 0.05; Table 4). Moreover, 2 h transportation increased the concentration of NOS in thigh muscle (*p* < 0.05), but there was no significant difference between group 1 and group 3 (*p* > 0.05).

### 3.4. Muscle Metabolites

Table 5 shows the concentration of MDA and LA in the breast and thigh muscle increased with increasing transportation time (*p* < 0.05). Transportation significantly affected also the concentrations of glycogen in breast and thigh muscles (*p* < 0.05). However, the glycogen contents of both breast and thigh muscles decreased with an increase in the transportation time (*p* < 0.05).

### 3.5. HSP70 mRNA Transcription Expression

Table 6 shows that neither 2 h nor 4 h transportation influenced HSP70 mRNA levels in the liver of broilers (*p* > 0.05).

## 4. Discussion

The stress associated with transport involves a number of factors or stressors that contribute to the impact of transportation on birds. The stressor least likely to have been experienced by birds while housed on the farm, was mainly the vibration and motion resulting from the movement of transportation. In conjunction with the friction, collision, hunger, and thirst caused by transportation, broilers likely experienced a series of adverse psychological and physiological reactions. In this study, we used a transportation simulator for different time periods, along with feed and water deprivation, and quantified the effect by measuring aspects of blood chemistry, muscle metabolism, and meat quality.

The changes in the composition of the blood revealed the physiological response of broilers to stress [24]. The thyroid hormones regulate the metabolism of protein, fat, and carbohydrates [25], affecting the energetic efficiency of cells. Numerous physiological and pathological stimuli influence thyroid hormone synthesis. The data of the 2 h transportation tests showed depressed levels of T_4_ and T_3_ concentrations which agree with those reported by Yu et al. [26]. This depression suggests that stress increases the release of corticosterone in the birds which would reduce the synthesis of T_4_ and T_3_, and simultaneously suppress peripheral tissue 5-deiodination enzyme activity and inhibit the conversion of T_4_ to T_3_. Moreover, the suppression of the hypothalamic-pituitary-thyroid axis reduces the synthesis and secretion of thyroid hormone and increases the rate of clearance and utilization of T_4_ and T_3_ [27]. The 2 h transportation stress also caused a decline in serum insulin concentration. Transportation stress can cause the excitation of the sympathetic nervous-adrenal medulla and reduce insulin secretion, leading to changes in glucose metabolism. A significant rise in blood glucose was observed in meat-type, yellow-feathered chickens during transport [3]. In this study, serum insulin levels were decreased by transportation stress, but serum glucose levels were not significantly changed, which may be related to the decreased glycogen levels in breast and thigh muscles. Serum T_3_, T_4_, and insulin concentrations recovered as the time of transportation (4 h transportation) increased, which suggests that the broilers adjusted their metabolism to homeostatic equilibrium in response to the motion stress. Moreover, there appeared to be little or no tissue damage from the stress, as no change was observed in the serum activities of LDH, GGT, AST, and CK.

Transport significantly affected 24 h drip loss of breast and thigh muscle. This result is in accordance with Yue et al. [3]. The drip loss_24 h_ from the breast and thigh muscle increased significantly as the transport time increased, and this result agrees with former findings, where the pectoralis muscle of broilers transported for 3 h showed the highest drip loss [28,29]. However, our results are inconsistent with the results of another study, where no significant variations were found in drip loss of the pectoralis muscle in transported birds [30]. This difference might be related to the temperature and transportation intensity of the broilers.

Postmortem pH decline is one of the most important events in the conversion of muscle to meat, due to its impact on meat texture, color, and water-holding capacity. The rate of pH decline is dependent on the activity of glycolytic enzymes just after death; the ultimate pH is determined by the initial glycogen reserves of the muscle. A low pH is associated with poor water-holding capacity and poor functionality [31]. In this study, 4 h transport significantly affected pH_24 h_ of both thigh and breast muscles, as shown in previous studies [32]. This might increase the likelihood of pale, soft, and exudative (PSE)-like meat or deterioration of meat.

Generally, myocardial and skeletal muscles are the tissues mostly affected by transportation stress [33]. The myocardium increases peripheral blood supply by accelerating heart rate, and skeletal muscle facilitates body tension in order to maintain balance during movement. The physiological changes induced in the muscle tissue by transportation can result in tissue damage [34]. However, the mitochondrial antioxidant defense system enzymes, such as SOD, GSH, and catalase can help prevent muscles damage by the stressors imposed by transport. It is well-known that SOD is a protective enzyme that selectively scavenges (O_2_^−^) by catalyzing its dismutation to H_2_O_2_ and molecular oxygen (O_2_). Likewise, glutathione peroxidase (GSH-Px) catalyzes the conversion of H_2_O_2_ to water by using reduced glutathione (GSH) as a cofactor [35]. The activity of these enzymes reflects the effects of transport on the avian antioxidant system and the ability of the bird to adapt to transportation. In this study, transportation stress did not affect the activities of GSH in both breast and thigh muscle. However, the activity of SOD in both muscles increased by 2 h transportation, and tended to recover after 4 h transportation, but was still higher compared with the control group. These results indicate that 2 h transportation increased the antioxidant activity of chickens, and it appeared that adaption occurred as the duration of the stress (for example 4 h transportation) was extended.

Malondialdehyde (MDA) is a product formed during lipid peroxidation and indirectly reflects the extent of cell and tissue damage [36]. This experiment confirmed that transportation stress increased MDA concentrations in breast and thigh muscles, suggesting that the stress induced cell damage and disruption of muscle tissue. It is expected that muscular damage will be more serious when transport is prolonged, as indicated by the increase in MDA in the breast and thigh muscle with the increase in the time of transportation. These findings are in agreement with those of Zhang et al. [37], who found that the muscle content of MDA increased in Cobb broilers transported for 3 h.

The decline of glycogen concentration in muscles with increasing transportation time suggests that liver glycogen breaks down rapidly when the stress commences [38]. The quantity of glycogen stored is limited and insufficient to maintain plasma glucose, as birds must also cope with the additional stress of feed withdrawal during transport. This result is consistent with Vosmerova [39] and Dadgar et al. [40]. In this experiment, the concentration of LA in both breast and thigh muscle increased with the duration of transportation. These results suggest that lactate accumulation caused by rapid anaerobic glycolysis is the main reason for the muscle pH decline of transported chickens [41]. However, Yue et al. found that transport time did not affect the concentration of lactate in either breast or thigh muscles of the meat-type yellow-feathered chickens [3]. Perhaps this inconsistency is associated with the different chicken genotypes used. In the present experiment, higher lactate content in both muscles were accompanied with pH decline and higher drip loss in 4 h transported broilers, suggesting that the enhancement of glycolysis and the accumulation of lactate were the main reasons of the transport induced detrimental changes of meat quality.

When animals are experiencing stress, the amount of HSPs in the body increases dramatically to protect organs and cells from damage. HSP70 mRNA concentrations in pig muscle increased following transport for 2 h, but appeared to recover after 4 h, but was increased by extending the time to 6 h [42]. Moreover, other reports in broilers show that transport up-regulated mRNA expression of HSP70 in both pectoralis major [43] and tibialis anterior muscles [44]. However, in our study, transportation did not influence the transcription level of HSP70 mRNA in the liver of broilers. Perhaps the degree (intensity and duration) of stress was insufficient, although heat stress alone can increase HSP70 transcription [45]. Moreover, it was also reported that the content of HSP70 did not change significantly during 0.5 h transport compared to the control group, but was significantly higher after 1 h or more of transport [30].

## 5. Conclusions

Simulation transport stress induced significant physiological changes in broiler chickens. Simulation of the vibration and motion associated transport for 2 h alters the secretion of hormones and the biochemical characteristic of blood. However, after 4 h, birds appear to regain homeostatic equilibrium. Despite this, the stress activated antioxidant defenses, acidified muscles, increased peroxidation, as well as decreased the water-holding capacity of muscle tissues. This study supports and extends previous work that identified transportation as a major risk in relation to bird welfare and meat quality. It provides a better understanding of the effect of transport duration on meat quality, and guidance for the exploration of effective measures to reduce the stress response and improve meat quality of transported broilers.

## Figures and Tables

**Figure 1 animals-10-01442-f001:**
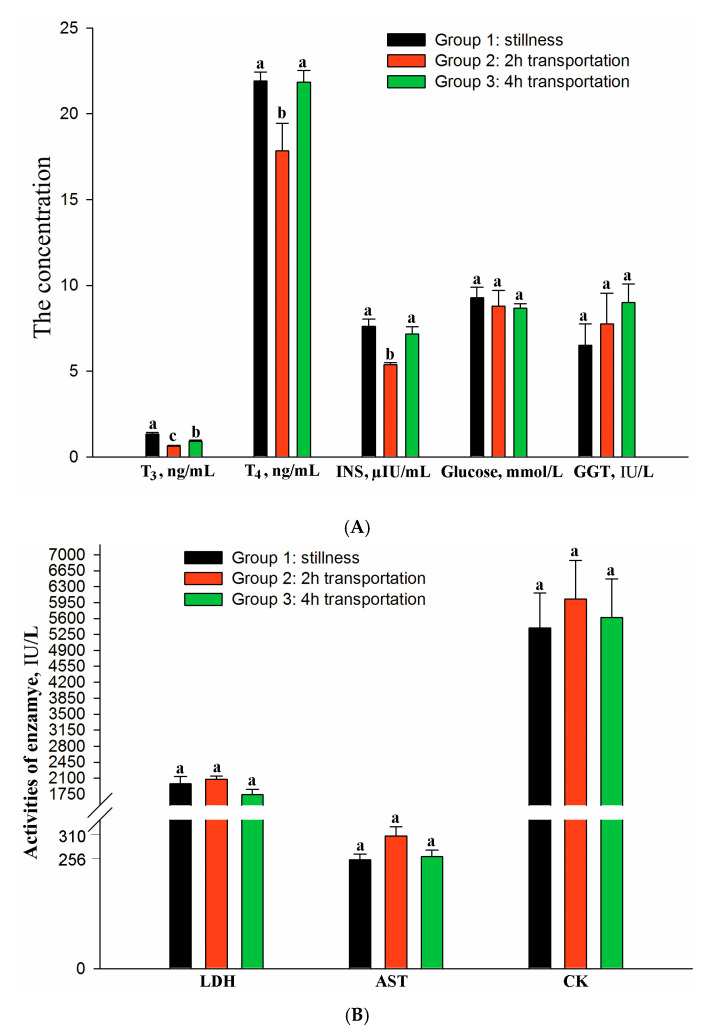
Effect of different transportation treatments on the blood hormones and metabolites of broilers. (**A**) changes of T_3_, T_4_, INS, Glucose, and GGT concentration in the serum and (**B**) changes of LDH, AST, and CK concentration in the serum. Bars within a cluster with different superscripts differ (*p* < 0.05). T_3_ = triiodothyronine, T_4_ = thyroxine, INS = insulin, GGT = γ-glutamyl transferase; LDH = lactate dehydrogenase, AST = asparate aminotransferase, and CK = creatine kinase.

**Figure 2 animals-10-01442-f002:**
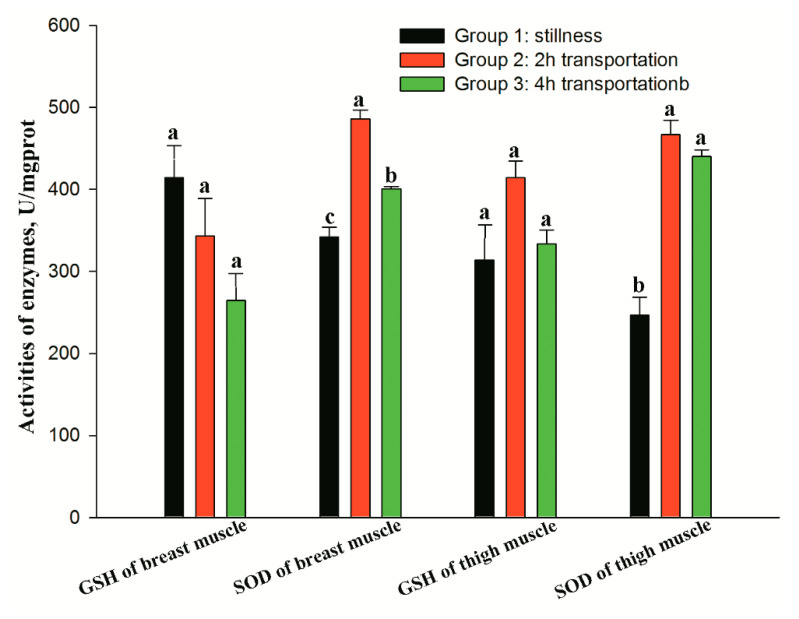
Effect of different transportation treatments on activities of GSH and SOD enzymes from breast and thigh muscle of broilers. Bars within a cluster with different superscripts differ (*p* < 0.05). GSH = glutathione peroxidase, and SOD = superoxide dismutase.

**Table 1 animals-10-01442-t001:** Composition (%) and nutrient levels of diets (air dry basis).

Ingredients	0–3 Week	4–6 Week
Maize	62.20	59.20
Soybean meal	29.20	33.26
Soybean oil	1.40	2.71
Fishmeal	3.00	1.00
Met	0.10	0.03
Lys	-	0.01
Dicalcium phosphate	1.50	1.33
Limestone	1.30	1.17
Sodium chloride	0.30	0.30
Vitamin–mineral premix ^1^	1.00	1.00
Nutrient levels ^2^		
ME(MJ/kg)	12.21	12.33
CP, %	21.00	20.00
Ca, %	1.00	0.90
Available *p*, %	0.45	0.43
Lys, %	1.12	1.00
Met + Cys, %	0.78	0.71

^1^ The premix provided the following per kg diet: vitamin A 10,000 IU, vitamin D_3_ 2000 IU, vitamin E 10 IU, vitamin K_3_ 2.5 mg, vitamin B_1_ 1 mg, vitamin B_2_ 6 mg, vitamin B_3_ 10 mg, vitamin B_5_ 40 mg, vitamin B_6_ 3 mg, vitamin B_11_ 0.3 mg, vitamin B_12_ 0.01 mg, biotin 0.12 mg, Cu (as copper sulfate) 8 mg, Fe (as ferrous sulfate) 80 mg, Mn (as manganese sulfate) 60 mg, Zn (as zinc sulfate) 40 mg, Se (as sodium selenite) 0.15 mg, and I (as potassium iodide) 0.35 mg. ^2^ Calculated values.

**Table 2 animals-10-01442-t002:** Primer sequences used for real-time PCR of HSP70 gene in the liver.

Gene	Primer Sequence (5′–3′)	PCR Product (bp)	Melting Temperature (°C)
HSP70	F: GCTTATGGTGCCGCTGTG R: TGGTGGTGTTACGCTTGATG	151	55.7
18SrRNA	F: GACACGGACAGGATTGACAG R: CCAGAGTCTCGTTCGTTATCG	120	55.6

**Table 3 animals-10-01442-t003:** Effect of different transportation treatments on drip loss and pH in broiler muscle.

Item	Treatment			*p*-Value	Pooled SE
	Stillness	2 h Transportation	4 h Transportation		
Breast muscle					
Drip loss_24_, %	0.54 ^a^	1.20 ^b^	1.47 ^c^	0.000	0.12
pH _45 min_	6.31 ^a^	6.25 ^a^	6.03 ^b^	0.039	0.05
pH _24 h_	5.99 ^a^	5.87 ^a^	5.31 ^b^	0.045	0.06
Thigh muscle					
Drip loss_24_, %	0.56 ^a^	1.11 ^b^	1.24 ^b^	0.000	0.09
pH _45 min_	6.21 ^a^	6.19 ^a^	6.02 ^b^	0.003	0.08
pH _24 h_	6.10 ^a^	6.16 ^a^	5.89 ^b^	0.002	0.05

^a–c^ Superscripts for means are for one-way ANOVA analyses. Within a column, means without a common superscript differ significantly (*p* < 0.05).

**Table 4 animals-10-01442-t004:** Effect of different transportation treatments on nitric oxide synthase (NOS) activity in broiler muscle.

Treatment	NOS (U/mg Prot)	
	in the Breast Muscle	in the Thigh Muscle
Stillness	0.67	0.48 ^a^
2 h transportation	0.61	0.64 ^b^
4 h transportation	0.70	0.38 ^a^
*p*-value	0.549	0.000
Pooled SE	0.03	0.03

^a,b^ Superscripts for means are for one-way ANOVA analyses. Within a column, means without a common superscript differ significantly (*p* < 0.05).

**Table 5 animals-10-01442-t005:** Effect of different transportation treatments on metabolites in broiler muscle.

Item	Treatment			*p*-Value	Pooled SE
	Stillness	2 h Transportation	4 h Transportation		
Breast muscle					
LA ^1^ (mmol/mg prot)	113.61 ^a^	121.09 ^b^	122.27 ^b^	0.008	2.71
MDA ^2^ (nmol/mg prot)	5.73 ^a^	6.22 ^b^	7.76 ^c^	0.000	0.25
Glycogen (mg/g)	5.97 ^b^	5.43 ^c^	4.16 ^a^	0.000	0.63
Thigh muscle					
LA (mmol/mg prot)	59.79 ^a^	64.20 ^b^	77.87 ^c^	0.000	2.29
MDA (nmol/mg prot)	3.41 ^a^	4.29 ^b^	4.96 ^c^	0.026	0.26
Glycogen (mg/g)	7.23 ^c^	5.53 ^b^	3.19 ^a^	0.000	0.38

^a–c^ Superscripts for means are for one-way ANOVA analyses. Within a column, means without a common superscript differ significantly (*p* < 0.05). ^1^ LA lactic Acid. ^2^ MDA = malondialdehyde.

**Table 6 animals-10-01442-t006:** Effect of different transportation treatments on the relative mRNA expression of HSP70 in broiler livers.

Treatment	The Relative Expression Level of HSP70 ^1^
Stillness	1.00
2 h transportation	0.96
4 h transportation	0.97
*p*-value	0.533
Pooled SE	0.03

^1^ HSP70 = Heat stress protein 70.

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
