# Peer review of "Stress Associated with Simulated Transport, Changes Serum Biochemistry, Postmortem Muscle Metabolism, and Meat Quality of Broilers"

_animals, 2020, doi:10.3390/ani10081442_

Round 1
Reviewer 1 Report
Line 17: „broilers chickens” should be changed to „broiler chickens”.
Line 38: „of the broiler” should be changed to „of the broilers”.
Line 42: „on the animal or bird”. I do not understand: birds are also animals.
Line 49: „Transportation combined food withdrawal and catching” should be changed to „Transportation combined with food withdrawal and catching”.
Line 57: „broilers chickens” should be changed to „broiler chickens”.
Line 64: „AA male chicks”. It should be indicated, what the abbreviation „AA” means.
Table 1: „Sodium cholride” should be changed to „Sodium chloride”.
Line 93: „was analyzed” should be changed to „were analyzed”.
Line 132: „in the Table 3” should be changed to „in Table 3”.
Line 136: The pH values are not indexes, therefore „these indexes did not significantly change” should be changed to „these values did not change significantly”.
Lines 142-143: The word „significantly” should be inserted into the sentence: „Transportation stress did not affect the activities of GSH in both breast and thigh muscle significantly (P>0.05)”.
Line 144: The reference to Figure 2 is earlier than the reference to Table 4 (Line 146). Therefore, Figure 2 should be moved before Table 4 in the text.
Lines 144-145: „this index was still higher that of the control group” should be changed to „these values were still higher than that of the control group”.
Lines 158-159: „Table 5 shows the concentration of MDA and LA in the breast and thigh muscle increased with the increase in the time of transportation (P<0.05)” should be changed to „Table 5 shows that the concentration of MDA and LA in the breast and thigh muscle increased with increasing the time of transportation (P<0.05)”.
Lines 159-160: „Transportation significantly affected the concentrations of glycogen in breast and thigh muscles (P<0.05)” should be changed to „Transportation significantly affected also the concentrations of glycogen in breast and thigh muscles (P<0.05)”.
Lines 160-161: „The glycogen in both breast and thigh muscles was reduced with increasing transportation times (P<0.05)” should be changed to „However, the glycogen content of both breast and thigh muscles decreased with increasing the transportation time (P<0.05)”.
Lines 202-203: „this result is in accordance with that the pectoralis muscle of 3 h transportation showed the higher drip loss” should be changed to „this result is in accordance with former findings, where the pectoralis muscle of 3 h transportation showed the higher drip loss”.
Lines 203-204: „But our results is inconsistent with that no significant variations were found in drip loss of pectoralis muscle in transport groups” should be changed to „However, our results are inconsistent with the results of another study, where no significant variations were found in the drip loss of pectoralis muscle in transport groups”.
Line 212: „affected most” should be changed to „mostly affected”.
Lines 214-215: „The physiological changes induced muscle tissue by transportation” should be changed to „The physiological changes induced in the muscle tissue by transportation”.
Line 217: „well-know” should be changed to „well-known”.
Line 223: „was increased” should be changed to „increased”.
Line 229: „imposed” should be deleted.
Lines 240-241: „Yue et al. found transport time did not affect” should be changed to „Yue et al. found that transport time did not affect”.
Line 245: „the main reasons that transport induced detrimental changes” should be changed to „the main reasons of the transport induced detrimental changes”.
References:
In some cases the full name, otherwise the abbreviation of the journals is used. The journal abbreviations are sometimes not correct (For example: Br Poult Sci is Brit Poultry Sci and Poult Sci is Poultry Sci officially).
The english grammar should be checked and corrected mainly in the „Discussion” chapter.
Author Response
Line 17: „broilers chickens” should be changed to „broiler chickens”.
Line 38: „of the broiler” should be changed to „of the broilers”.
Line 42: „on the animal or bird”. I do not understand: birds are also animals.
Line 49: „Transportation combined food withdrawal and catching” should be changed to „Transportation combined with food withdrawal and catching”.
Line 57: „broilers chickens” should be changed to „broiler chickens”.
Line 64: „AA male chicks”. It should be indicated, what the abbreviation „AA” means.
Table 1: „Sodium cholride” should be changed to „Sodium chloride”.
Line 93: „was analyzed” should be changed to „were analyzed”.
Line 132: „in the Table 3” should be changed to „in Table 3”.
Line 136: The pH values are not indexes, therefore „these indexes did not significantly change” should be changed to „these values did not change significantly”.
Lines 142-143: The word „significantly” should be inserted into the sentence: „Transportation stress did not affect the activities of GSH in both breast and thigh muscle significantly (P>0.05)”.
Line 144: The reference to Figure 2 is earlier than the reference to Table 4 (Line 146). Therefore, Figure 2 should be moved before Table 4 in the text.
Lines 144-145: „this index was still higher that of the control group” should be changed to „these values were still higher than that of the control group”.
Lines 158-159: „Table 5 shows the concentration of MDA and LA in the breast and thigh muscle increased with the increase in the time of transportation (P<0.05)” should be changed to „Table 5 shows that the concentration of MDA and LA in the breast and thigh muscle increased with increasing the time of transportation (P<0.05)”.
Lines 159-160: „Transportation significantly affected the concentrations of glycogen in breast and thigh muscles (P<0.05)” should be changed to „Transportation significantly affected also the concentrations of glycogen in breast and thigh muscles (P<0.05)”.
Lines 160-161: „The glycogen in both breast and thigh muscles was reduced with increasing transportation times (P<0.05)” should be changed to „However, the glycogen content of both breast and thigh muscles decreased with increasing the transportation time (P<0.05)”.
Lines 202-203: „this result is in accordance with that the pectoralis muscle of 3 h transportation showed the higher drip loss” should be changed to „this result is in accordance with former findings, where the pectoralis muscle of 3 h transportation showed the higher drip loss”.
Lines 203-204: „But our results is inconsistent with that no significant variations were found in drip loss of pectoralis muscle in transport groups” should be changed to „However, our results are inconsistent with the results of another study, where no significant variations were found in the drip loss of pectoralis muscle in transport groups”.
Line 212: „affected most” should be changed to „mostly affected”.
Lines 214-215: „The physiological changes induced muscle tissue by transportation” should be changed to „The physiological changes induced in the muscle tissue by transportation”.
Line 217: „well-know” should be changed to „well-known”.
Line 223: „was increased” should be changed to „increased”.
Line 229: „imposed” should be deleted.
Lines 240-241: „Yue et al. found transport time did not affect” should be changed to „Yue et al. found that transport time did not affect”.
Line 245: „the main reasons that transport induced detrimental changes” should be changed to „the main reasons of the transport induced detrimental changes”.
We have revised and complemented corresponding contents according to the expert’s suggestion. Please see the revised version for details.
References:
In some cases the full name, otherwise the abbreviation of the journals is used. The journal abbreviations are sometimes not correct (For example: Br Poult Sci is Brit Poultry Sci and Poult Sci is Poultry Sci officially).
The full name of the journals format is used in the references and the wrong journal abbreviations have been corrected. Please see the revised version for details.
The english grammar should be checked and corrected mainly in the „Discussion” chapter.
We have checked and corrected the grammar of the paper. Please see the revised version for details.
Reviewer 2 Report
The manuscript is interesting and it should be published after minor revisions. I made my suggestions in the attached text.

Author Response
- It would be nice to know more details in the rearing aspects of the chicks, flock density, and some characteristics of the housing. (Line 66)
Response: We have supplemented. “Birds were raised in accordance with the AA Broiler Management Guide. Chicks were vaccinated for Marek’s Disease at 1 day-old and for Newcastle Disease and Infectious Bronchitis at 7 days post-hatching. Room temperature was maintained at 33°C for days 0-3 and gradually reduced to 24°C and maintained at 24℃ till the end of the study. Photoperiod was controlled to 16 hours of light and 8 hour of darkness. Relative humidity was set at 60%-70% during the first week and then at 50%-60% for the rest of the experiment.” Please see Line 75-81.
- 100 rpm is surely a very low speed transportation. State which is the normal speed for transporting broilers.
Response: “the transportation vibration rate was set at 100 rpm” 100 rpm is mostly equal to the speed of 60km/ h on the country road in China.
- Change “2h” to “2 h”
Response: We have changed “2h” to “2 h”.
- State the found p value.
Response: We have added P value for the parameters presenting in histograms. Other parameters were shown in the table.
- Delete “ Significantly”
Response: We have deleted them.
- Your results are quite different from the expected, since exposure to stress tends to increase the glucose in blood. I believe you should provide a better discussion than simply rely on the bird strain. Line 193
Response: We have discussed this as “In this study, serum insulin levels were decreased by transportation stress, but serum glucose levels were not significantly changed, which may be related to the decreased glycogen levels in breast and thigh muscles.”
- The sentence is not clear, please consider rewriting it. Line 203
Response: We have rewritten. “However, our results are inconsistent with the results of another study, where no significant variations were found in drip loss of the pectoralis muscle in transported birds”
- I think it would be helpful if you add the time broilers are normally transported to slaughter in some countries to help the reader understand better your points. Line 225
Response: We have added transportation time. “These results indicate that 2 h transportation increased the antioxidant activity of chickens and it appeared that adaption occurred as the duration of the stress (for example 4 h transportation) was extended.”
- You should discuss more this finding. Line 252
Response: We have discussed as “Perhaps the degree (intensity and duration) of stress was insufficient, although heat stress alone can increase HSP70 transcription.”
- Do you have a reference to support that? Line 255
Response: We have added a reference here. “Mishra Aditya, Patel Pragati, Singh HS, Goyal Girraj, Baghel RPS, Pankaj PK: Effect of ascorbic acid supplementation on mRNA expression of HSP70 gene in broilers exposed to heat stress. Indian Journal of Poultry Science 2015, 50(3):282-287.”
Reviewer 3 Report
The manuscript was properly conducted and findings reported are important for poultry production and health. The paper contains important data health of chickens under different dietary treatments. The Authors investigated an interesting topic and the objective of the paper is of worldwide interest and fits well within the overall scope of the journal. Results were properly reported and the findings have been accurately discussed and compared with other recently published papers.
My only suggestion is that you change the labels on all tables, from Group 1, 2, and 3, to the actual name of the group, as you have it in the figures.
Author Response
My only suggestion is that you change the labels on all tables, from Group 1, 2, and 3, to the actual name of the group, as you have it in the figures.
We have changed the labels according to the reviewer’s suggestion. Changed Group 1,2,3 to stillness, 2 h transportation and 4 h transportation.
Reviewer 4 Report
This paper describes the use of a transport simulation to investigate the effects of the stress associated with transport on various blood biochemical parameters in broilers. Overall, the study seems interesting, but it lacks quite a lot of detail in many sections of the paper. Improvements that could be made to each section are discussed below with specific points at the end of the document.
Title: It is not immediately obvious that the transport was simulated. Maybe try to include the term “simulated transport” in the title.
Simple Summary: Could be simplified in parts, e.g. “endogenous contamination” is not an accessible term for a lay audience. The guide for authors states that “the simple summary consists of no more than 200 words in one paragraph and contains a clear statement of the problem addressed, the aims and objectives, pertinent results, conclusions from the study and how they will be valuable to society. This should be written for a lay audience"
Abstract: The abstract starts with results. The guide for authors states the abstract should follow the style of structured abstracts i.e. Background; 2) Methods; 3) Results; and 4) Conclusion
Introduction: The introduction is very short and needs more detail. The authors should explicitly state why this experiment was necessary and what new information will be gained. They should also discuss the use of the simulation model more clearly. The authors acknowledge that lots of different factors can cause stress during transport. Why is this model a good way to investigate this problem as opposed to actual transport?
Materials and Methods: Bird management and experimental design needs much more detail. There is no description of the size of the pens used to house the birds. The authors need to add much more detail as to how the birds were exposed to the simulation, were birds handled, or was the whole pen subjected to the transport simulation? A photograph/video of the chickens in the simulator, although not essential, could be a really nice addition to this paper.
Discussion: What are the implications of the study? What have we learned and what do they recommend in terms of reducing stress or studies to investigate limiting the impact of stress?
Specific comments
Line 16: Again, it is unclear that the transport was simulated. I would suggest something like “The objective of this study was to investigate the effect of transport stress on broilers through the use of a transport vibration and motion simulation machine.
Line 49: confusing
Line 65: Some more detail on the temperature and lighting would be useful
Line 82-88: How many birds were blood sampled per treatment? How was the muscle sampling carried out and again, how many samples were taken from how many birds?
Line 109-112: Blood biochemistry variables are often not normally distributed. Were the variables checked for outliers and normality? What terms were included in the ANOVA model?
Line 137: The alignment of numbers in the columns in Table 3 needs to be fixed.
Line 175/176: Is this a justification for looking into vibration and motion specifically? Needs more detail.
Line 177: Novel stressor for the birds or novel to study?
Line 179: Sounds like results from this paper as opposed to a reference to another, please reword.
Line 189: Please add reference.
Line 258: Short term transport stress “simulation”
Author Response
Title: It is not immediately obvious that the transport was simulated. Maybe try to include the term “simulated transport” in the title.
Response:The title has been revised to “Stress Associated with Simulated Transport Changes Serum Biochemistry, Postmortem Muscle Metabolism and Meat Quality of Broilers”
Simple Summary: Could be simplified in parts, e.g. “endogenous contamination” is not an accessible term for a lay audience. The guide for authors states that “the simple summary consists of no more than 200 words in one paragraph and contains a clear statement of the problem addressed, the aims and objectives, pertinent results, conclusions from the study and how they will be valuable to society. This should be written for a lay audience"
Response: We have rewritten the simple summary. “The objective of this study was to investigate the effects of the transportation on broilers through the use of a vibration and motion simulation machine. Short-term transport stress induced significant physiological changes in broiler chickens. Simulation of the vibration and motion associated transport for 2 h altered hormones secretion and blood biochemical characteristics. However, after 4h, birds appear to regain homeostatic equilibrium. Despite this, the stress activated antioxidant defenses, acidified muscles and increased peroxidation as well as decreasing the meat quality of broilers. This study supports and extends previous work that identified transportation as a major risk in relation to bird welfare and meat quality.”
Abstract: The abstract starts with results. The guide for authors states the abstract should follow the style of structured abstracts i.e. Background; 2) Methods; 3) Results; and 4) Conclusion
Response: We have complemented “Backgroud” parts in the abstract. Line 26-28 and 42-43.
Introduction: The introduction is very short and needs more detail. The authors should explicitly state why this experiment was necessary and what new information will be gained. They should also discuss the use of the simulation model more clearly. The authors acknowledge that lots of different factors can cause stress during transport. Why is this model a good way to investigate this problem as opposed to actual transport?
Response: We have complemented this content. “In an attempt to investigate the effect of these factors, a transport simulator was used in this study. The simulation model not only simulates different parameters of actual transportation, but also allows control of experimental repeatability.” Line 55-56, line 61-63.
Materials and Methods: Bird management and experimental design needs much more detail. There is no description of the size of the pens used to house the birds. The authors need to add much more detail as to how the birds were exposed to the simulation, were birds handled, or was the whole pen subjected to the transport simulation? A photograph/video of the chickens in the simulator, although not essential, could be a really nice addition to this paper.
Response: We feel very sorry because no photograph or video of the chickens in the simulator, but we reword this part in details. “Birds were raised in accordance with the AA Broiler Management Guide. Chicks were vaccinated for Marek’s Disease at 1 day-old and for Newcastle Disease and Infectious Bronchitis at 7 days post-hatching. Room temperature was maintained at 33°C for days 0-3 and gradually reduced to 24°C and maintained at 24℃ till the end of the study. Photoperiod was controlled to 16 hours of light and 8 hour of darkness. Relative humidity was set at 60%-70% during the first week and then at 50%-60% for the rest of the experiment.” and “The same person handled chickens in the same way and put them in the cage. Chickens were housed in cages, with a density of 8 birds/0.64m2. The whole cages (0.64 m2) were subjected to the transport simulation.” And line 100-109.
Discussion: What are the implications of the study? What have we learned and what do they recommend in terms of reducing stress or studies to investigate limiting the impact of stress?
Response: We have supplemented this part. “In conjunction with the friction, collision, hunger and thirst caused by transportation, broilers likely experience a series of adverse psychological and physiological reactions.” And “It provides a better understanding of the effect of transport duration on meat quality and guidance for the exploration of effective measures to reduce the stress response and improve meat quality of transported broilers.”
Specific comments
Line 16: Again, it is unclear that the transport was simulated. I would suggest something like “The objective of this study was to investigate the effect of transport stress on broilers through the use of a transport vibration and motion simulation machine.
Response: We have changed according to the reviewer’s suggestion. Line 15-17.
Line 49: confusing
Response: Change to “Transportation, combined with feed withdrawal and catching, altered the expression of hepatic genes associated with nutrient metabolism, cellular control and immune function of broiler chickens.”
Line 65: Some more detail on the temperature and lighting would be useful
Response: We have supplemented. “Birds were raised in accordance with the AA Broiler Management Guide. Chicks were vaccinated for Marek’s Disease at 1 day-old and for Newcastle Disease and Infectious Bronchitis at 7 days post-hatching. Room temperature was maintained at 33°C for days 0-3 and gradually reduced to 24°C and maintained at 24℃ till the end of the study. Photoperiod was controlled to 16 hours of light and 8 hour of darkness. Relative humidity was set at 60%-70% during the first week and then at 50%-60% for the rest of the experiment.” Please see Line 75-81.
Line 82-88: How many birds were blood sampled per treatment? How was the muscle sampling carried out and again, how many samples were taken from how many birds?
Response: We have supplemented. “Three birds of each replicate were randomly selected and weighed. Blood was collected by cardiac puncture using heparinized tubes. The serum was obtained by centrifugation at 3,000 × g at 4℃ for 5 min. The serum samples were frozen immediately in liquid nitrogen and stored at −80°C for the determination of serum parameters. One bird of each replicate was then euthanatized after electrical stunning [5]. Muscles of the breast and thigh were removed immediately and evaluated for pH (three different positions each bird) and drip loss (three pieces each bird), as described previously with modifications [16]. Three small pieces of breast muscle, leg muscle and liver of each bird were taken and frozen immediately in liquid nitrogen for biochemical determination.”
Line 109-112: Blood biochemistry variables are often not normally distributed. Were the variables checked for outliers and normality? What terms were included in the ANOVA model?
Response: We have supplemented. “Blood biochemical data were tested whether they were normal distribution. If not, the data would be transformed into normal distribution data, and then statistical analysis were conducted.”
Line 137: The alignment of numbers in the columns in Table 3 needs to be fixed.
Response: We have corrected it.
Line 175/176: Is this a justification for looking into vibration and motion specifically? Needs more detail.
Response: We changed to “The stressor less likely to have been experienced by birds while housed on the farm, is mainly the vibration and motion resulting from the movement of transportation.”
Line 177: Novel stressor for the birds or novel to study?
Response: We changed to “we used transportation simulator for different time periods, along with feed and water deprivation, and quantified the effect by measuring aspects of blood chemistry, muscle metabolism and meat quality.”
Line 179: Sounds like results from this paper as opposed to a reference to another, please reword.
Response: “This might indicate differences in the transport stress applied or the strain of bird used.”
Line 189: Please add reference.
Response: “Aguilera G: Chapter 8 - The Hypothalamic–Pituitary–Adrenal Axis and Neuroendocrine Responses to Stress. In: Handbook of Neuroendocrinology. Edited by Fink G, Pfaff DW, Levine JE. San Diego: Academic Press; 2012: 175-196.”
Line 258: Short term transport stress “simulation”
Response: We have corrected it.
Round 2
Reviewer 4 Report
The authors have addressed some of the changes requested adequately, however a number of important queries have been left unanswered or were addressed poorly.
Introduction
The introduction is still quite short. The simulation model is still not described in the introduction and there is no specific reference to other authors who have used similar simulation models and to what their specific findings were. What specific improvements on previous work do the authors hope to discover?
Bird management and experimental design
How were the birds housed before they were put into treatment groups? Were they at a similar stocking density for their whole lives or was there a change in stocking density when the birds were put into treatment groups and exposed to the simulation?
Statistical analysis
The authors response to the statistical analysis query did not fill me with confidence. The authors still have not mentioned if or how they interrogated the data for outliers. The authors now state that if the data were not normal, a transformation was performed. How did the authors determine normality? What variables were determined non-normal and what transformations were used? The authors have still failed to mention what factors were included in the ANOVA models. There is not enough information to determine if the statistics were conducted correctly, or for another author to repeat the same analysis.
Alignment of Tables
The alignment of the numbers has not been corrected and the general alignment in some tables is now incorrect.
Highlighting of changes
These often do not always match up with the lines quoted by the authors in their response.
Author Response
Introduction
The introduction is still quite short. The simulation model is still not described in the introduction and there is no specific reference to other authors who have used similar simulation models and to what their specific findings were. What specific improvements on previous work do the authors hope to discover?
Response: We have complemented as “Two different levels of vibration for four intervals of time (0, 30, 60 and 90 minutes) were subjected to simulate the transportation using mechanical simulator, which did not affect chicken rectal temperature and weight loss [16]. Moreover, increasing vibration duration elevated the total white blood cell count, and heterophil-to-lymphocyte ratio of the hens. Simulated transportation did not affect egg production and plasma protein, but had an effect on the hens’ feed intake and blood glucose level [17]. Most of the studies investigated the effects of actual road transport on broilers [18-20], while there are few studies on the effects of simulated transport on the meat quality of pre-slaughter broilers.”
Bird management and experimental design
How were the birds housed before they were put into treatment groups? Were they at a similar stocking density for their whole lives or was there a change in stocking density when the birds were put into treatment groups and exposed to the simulation?
Response: The stocking density is different between in the house and in the simulator. We have complemented as “The birds were housed in the Nankou experimental farm of the Feed Research Institute, CAAS, Beijing, China. The stocking density is 8 birds /m2 in the cage.”
Statistical analysis
The authors response to the statistical analysis query did not fill me with confidence. The authors still have not mentioned if or how they interrogated the data for outliers. The authors now state that if the data were not normal, a transformation was performed. How did the authors determine normality? What variables were determined non-normal and what transformations were used? The authors have still failed to mention what factors were included in the ANOVA models. There is not enough information to determine if the statistics were conducted correctly, or for another author to repeat the same analysis.
Response: We have complemented as “Outliers were determined as values that deviated from the treatment mean by more than 1.5 times the interquantile range. (Devore, J., and R. Peck. 1993. Statistics: The exploration and analysis of data. 2nd ed. Duxbury Press, Belmont, CA.). No outliers were identified and removed from the data set. The normality of variables was tested using the Explore procedure. The variance of INS was non-normal and the transformation of ARSIN (SQRT(INS/10)) was used. The homogeneity of the variance among treatments was performed. In the ANOVA models, transportation condition is the main effect (factor).”
Alignment of Tables
The alignment of the numbers has not been corrected and the general alignment in some tables is now incorrect.
Response: The alignment of the numbers has been corrected in Table 3, Table 4 and Table 5.
Highlighting of changes
These often do not always match up with the lines quoted by the authors in their response.
Response: The reason for this inconsistence is that I used the line number of the track changes format, while the reviewer might get the clean edition.